# Thymoquinone Enhances Paclitaxel Anti-Breast Cancer Activity via Inhibiting Tumor-Associated Stem Cells Despite Apparent Mathematical Antagonism

**DOI:** 10.3390/molecules25020426

**Published:** 2020-01-20

**Authors:** Hanan A. Bashmail, Aliaa A. Alamoudi, Abdulwahab Noorwali, Gehan A. Hegazy, Ghada M. Ajabnoor, Ahmed M. Al-Abd

**Affiliations:** 1Department of Clinical Biochemistry, Faculty of Medicine, King Abdulaziz University, Jeddah 21589, Saudi Arabia; hanan.a.bashmail@gmail.com (H.A.B.); aliaa.alamo@gmail.com (A.A.A.); wal5566@gmail.com (A.N.); gehanhegazy@hotmail.com (G.A.H.); ga_clinbio@yahoo.com (G.M.A.); 2Department of Medical Biochemistry, Medical Division, National Research Centre, Giza 12622, Egypt; 3Department of Pharmaceutical Sciences, College of Pharmacy, Gulf Medical University, Ajman 4184, UAE; 4Department of Pharmacology, Medical Division, National Research Centre, Giza 12622, Egypt

**Keywords:** paclitaxel, thymoquinone, apoptosis, autophagy, tumor-associated stem cells

## Abstract

Thymoquinone (TQ) has shown substantial evidence for its anticancer effects. Using human breast cancer cells, we evaluated the chemomodulatory effect of TQ on paclitaxel (PTX). TQ showed weak cytotoxic properties against MCF-7 and T47D breast cancer cells with IC_50_ values of 64.93 ± 14 µM and 165 ± 2 µM, respectively. Combining TQ with PTX showed apparent antagonism, increasing the IC_50_ values of PTX from 0.2 ± 0.07 µM to 0.7 ± 0.01 µM and from 0.1 ± 0.01 µM to 0.15 ± 0.02 µM in MCF-7 and T47D cells, respectively. Combination index analysis showed antagonism in both cell lines with CI values of 4.6 and 1.6, respectively. However, resistance fractions to PTX within MCF-7 and T47D cells (42.3 ± 1.4% and 41.9 ± 1.1%, respectively) were completely depleted by combination with TQ. TQ minimally affected the cell cycle, with moderate accumulation of cells in the S-phase. However, a significant increase in Pre-G phase cells was observed due to PTX alone and PTX combination with TQ. To dissect this increase in the Pre-G phase, apoptosis, necrosis, and autophagy were assessed by flowcytometry. TQ significantly increased the percent of apoptotic/necrotic cell death in T47D cells after combination with paclitaxel. On the other hand, TQ significantly induced autophagy in MCF-7 cells. Furthermore, TQ was found to significantly decrease breast cancer-associated stem cell clone (CD44+/CD24-cell) in both MCF-7 and T47D cells. This was mirrored by the downregulation of TWIST-1 gene and overexpression of SNAIL-1 and SNAIL-2 genes. TQ therefore possesses potential chemomodulatory effects to PTX when studied in breast cancer cells via enhancing PTX induced cell death including autophagy. In addition, TQ depletes breast cancer-associated stem cells and sensitizes breast cancer cells to PTX killing effects.

## 1. Introduction

Over the past three decades, 1355 new drugs were approved for the treatment of malignancies [1,2]. However, there are 18.1 million new cases of cancer, and 9.6 million mortalities due to cancer annually [3]. Breast cancer has the highest incidence, causing the most female mortalities among other malignancies [4]. Breast cancer tissue is a heterogeneous tissue consisting of various cell types, which differ in terms of origin, function, genetic profile, morphology, and sensitivity to therapy [5,6]. Breast cancer stem cells (BCSCs) are a subclone of cancer cells that have gained great attention, and are believed to be responsible for tumor growth and unlimited self-renewal ability. BCSCs possess a remarkable ability to effectively persist after exposure to chemotherapy [7]. 

Paclitaxel (PTX) is anti-microtubule chemotherapy that has been used successfully for different types of solid tumors including breast cancer for more than 40 years [8,9,10]. PTX stabilizes tubulin dimmers and suppresses microtubule depolymerization during mitosis, resulting in cell cycle arrest in M-phase; it is also called mitotic catastrophe [9,11]. However, breast cancer patients treated with taxane frequently develop chemotherapeutic resistance [12]. Combination therapy for PTX has been studied by many research teams, including ours, to enhance its anti-tumor activity and protect PTX from tumor resistance [13,14,15,16]. 

Natural compounds are believed to be a promising alternative for many chemotherapeutic remedies in fighting neoplasia [17]. More than 74% of the newly approved anticancer drugs during the past 30 years were of natural origin or inspired by natural product [1,2]. *Nigella sativa* and its constituents are among the most studied medicinal herbs in different health care issues [18]. Thymoquinone (TQ) is the major natural component of *Nigella sativa* seeds; it possesses anti-bacterial, anti-oxidant, anti-allergic, and anti-cancer effects [19,20,21,22]. 

Medicinal plants combined with cancer chemotherapy has gained great attention in recent years, and some studies have demonstrated promising results and outcomes. The main goal of these studies was to reduce the chemotherapeutic resistance associated with conventional chemotherapeutic agents or to protect normal tissues from their toxicity [23]. In our previous publications, thymoquinone was shown to improve the activity of cisplatin and gemcitabine against head and neck squamous cell carcinoma and breast cancer cells in addition to protecting oral epithelial cells from cisplatin-induced apoptosis. Herein, we studied the effect of TQ on the cytotoxicity profile of PTX against breast cancer cells, emphasizing breast-cancer-resistant clones in relation to BCSCs.

## 2. Results

### 2.1. The Chemomodulatory Effect of Thymoquinone to PTX within Breast Cancer Cells

A sulfarodamine-B (SRB) assay was used to assess the effect of TQ on the cytotoxic profile of PTX against breast cancer cells by calculating the IC_50_ values and R-fractions of single and combined PTX against MCF-7 and T47D cells. PTX showed a dose-dependent cytotoxic effect. Viability started to drop significantly at a concentration of 0.1 μM with IC_50_ values of 0.2 ± 0.07 μM and 0.1 ± 0.01 μM in MCF-7 and T47D cells, respectively (Figure 1A,B). In contrary, TQ did not exert any cytotoxic activity against either cell line until 30 μM. Higher concentrations of TQ induced a sudden drop in the viability with calculated IC_50_ values of 64.9 ± 14 μM and 165.1 ± 2.8 μM in MCF-7 and T47D cells, respectively (Figure 1A,B). Equitoxic combination (100:1) of TQ with PTX did not further improve the IC_50_ values of PTX against either MCF-7 or T47D cells (0.7 ± 0.01 μM and 0.15 ± 0.02 µM, respectively). Combination index analysis showed that TQ antagonized the cell-killing effect of PTX against MCF-7 and T47D cells, resulting in CI-values of 4.6 and 1.6, respectively (Table 1). Yet, TQ completely abolished the resistance fractions of both MCF-7 and T47D towards PTX from 42.37 ± 1.4% and 41.9 ± 1.1%, respectively, to 0% (Figure 1A,B) (Table 1). These data suggest that TQ does not improve PTX potency against MCF-7 or T47D cells and apparently antagonizes its killing effects. However, TQ significantly abolishes tumor-associated resistant cell clones. 

### 2.2. Cell Cycle Distribution Analysis of Breast Cancer Cells

Further assessment for the interaction between TQ and PTX against cell cycle progression was undertaken using DNA content flow cytometry. In MCF-7 cells, PTX significantly arrested the cell cycle at G2/M-phase with a significant increase in the G2/M-phase population from 17.5 ± 2.3% to 71.2 ± 0.8% and from 15.9 ± 2% to 72.1 ± 2.8% after 24 h and 48 h, respectively (Figure 2A,B). TQ alone did not cause any significant change in the cell cycle distribution of MCF-7 cells. However, a combination of TQ with PTX induced a significant increase in the S-phase cell population (from 19.9 ± 0.5% to 23.8 ± 1%) after 24 h (Figure 2A). The cell cycle arrest at G2/M-phase induced by PTX alone and in combination with TQ resulted in cell death; a significant increase of Pre-G phase population was observed from 4.7 ± 1.5% to 27.1 ± 5% and 29.8 ± 4%, respectively, after 24 h (Figure 2C) and from 2.5 ± 0.6% to 17.9 ± 1.6%, 18.9 ± 0.4%, respectively, after 48 h (Figure 2D).

Similar to MCF-7, PTX significantly arrested T47D cells in G2/M-phase with a significant increase in this population from 19.4 ± 1.7% to 62.0 ± 2.9% and from 16.6 ± 1% to 83.3 ± 2.1% after 24 h and 48 h, respectively (Figure 3A,B). After 48 h of exposure, TQ alone and TQ+PTX treatment increased the S-phase T47D cell population from 29.1 ± 1.7% to 38.4 ± 0.2% and from 15.1 ± 1.7% to 28.6 ± 4.1%, respectively (Figure 3B). Interestingly, TQ treatment alone induced significant cell death and increased the Pre-G cell population of T47D cells from 8.5 ± 0.3% to 10.8 ± 0.2% and from 12.6 ± 1.4% to 67.6 ± 5.2% after 24 h and 48 h, respectively (Figure 3C,D). In addition, PTX alone induced a significant increase in the pre-G cell population from 8.5 ± 0.3% to 38.1 ± 5.1% and 12.6 ± 1.45% to 44.5 ± 3.2% after 24 and 48 h, respectively. Combination of PTX with TQ resulted in a significantly higher pre-G cell population compared to PTX treatment alone after 24 and 48 h (69.6 ± 1.1% and 60.4 ± 1.7%, respectively) (Figure 3C,D). Pre-G phase is indicative of cell death. However, it is non-specific and could be programmed cell death (apoptosis or autophagy) or non-programmed cell death (necrosis).

### 2.3. Apoptosis Assessment 

Herein, we investigated the effect of TQ in overcoming MCF-7, T47D cells resistance to PTX by inducing further apoptosis, necrosis, and/or autophagy. T47D cells were exposed to the pre-determined IC_50_ values of PTX, TQ, and their combination for 24 and 48 h rather than 72 h to detect early apoptotic events. Apoptosis/necrosis populations were then determined by Annexin-V/FITC-PI staining coupled with a flowcytometry technique. Both TQ alone and PTX alone induced a significant apoptosis after 24 h of exposure (22.4 ± 3.2% and 10.3 ± 0.8%, respectively) compared to untreated control T47D cells (4.3 ± 0.5%). Yet, PTX combination with TQ induced significantly more apoptosis compared to PTX treatment alone (58.1 ± 2.1%). In addition to apoptosis, TQ, PTX, and their combination induced significant necrotic cell death in T47D by 5.9 ± 0.3%, 7.4 ± 0.7%, and 22.2 ± 0.6%, respectively (compared to 2.3 ± 0.7% necrosis in control cells) (Figure 4A)). Similarly, further exposure (48 h) of T47D cells to PTX alone or TQ alone resulted in more apoptosis (18.4 ± 2.9% and 32.5 ± 2.8%, respectively) compared to untreated control cells (1.9 ± 0.5%). Combination of PTX and TQ did not significantly increase T47D apoptotic cell population compared to TQ treatment alone (31.5 ± 1.3%). Moreover, TQ, PTX, and their combination induced significant necrotic cell death in T47D by 4.0 ± 0.4%, 11.3 ± 0.6%, and 11.0 ± 0.4%, respectively (compared to 2.6 ± 0.1% necrosis in control cells) (Figure 4B). To confirm apoptosis, Western blot analysis was carried out for caspase-3 and PARP proteins. PTX induced the expression of caspase-3 after 24 and 48 h. Further combination of PTX with TQ resulted in more active caspase-3 and more cleavage for its downstream target protein, PARP (Figure 4C,D). 

### 2.4. Autophagy Assessment 

Besides apoptosis, we were keen to study the effect of PTX, TQ, and their combination on other cell death mechanisms such as the autophagy process. In MCF-7, treatment with PTX, TQ, and the combination of PTX+TQ increased the fluorescent intensity indicative of autophagic cell death by 58.2%, 33.9%, and 49.1%, respectively (Figure 5A). On the other hand, none of the treatments under investigation (PTX, TQ, or their combination) induced any significant change or autophagic cell death in T47D (Figure 5B). CQ (positive control autophagic drug) induced autophagic cell response in MCF-7 and T47D cell lines and increased Cyto-ID fluorescence by 31 and 42%, respectively (Figure 5A,B). For further conformation, two key autophagy genes beclin-1 and LC3-II were assessed by the RT-PCR technique. In MCF-7 cells, treatment with PTX, TQ, and the a combination of PTX+TQ significantly increased the expression of beclin-1 by 4.4, 3.1, and 6.8 folds, respectively, and significantly increased the expression of LC3-II by 3.4, 1.9, and 4.1 folds, respectively. In T47D, only PTX marginally increased the expression of beclin-1 by 1.4 folds (Figure 5C,D) 

### 2.5. Breast Cancer-Associated Stem Cell (CD44+/CD24- Cell Clone) Detection

Furthermore, we assessed the breast cancer-associated stem cell clone (CD44+/CD24-) and endothelial mesenchymal transition gene expression in relation to treatment with TQ, PTX, and their combination. TQ alone induced a significant decrease in the CD44+/CD24- stem cell clone by 12.4 ± 0.8%. However, PTX treatment reduced CD44+/CD24- cell clone by only 7.6 ± 0.1%. Interestingly, the combination of PTX with TQ significantly abolished the tumor-associated stem cell clone (CD44+/CD24-) by 32.3 ± 0.08% (Figure 6A). In addition, TQ significantly decreased the T47D associated stem cell clone (CD44+/CD24) by 19.9 ± 0.8%, while PTX caused a 9.9 ± 0.2% decrease in the tumor-associated stem cell clone. Yet, the combination of PTX with TQ further decreased the tumor-associated stem cell clone by 23.9 ± 1.6% (Figure 6B).

### 2.6. EMT Genes Expression Assessment 

Further assessment for the expression of key EMT genes (ZEB-2, TWIST-1, SNAIL-1, and SNAIL-2) after treatment with TQ and PTX was undertaken using the RT-PCR technique. No significant changes in the expression of ZEB-2 due to the treatment of TQ or PTX could be detected. TQ induced a significant increase in the expression level of both SNAIL-1 and SNAL-2 compared to the untreated control by 3.8 and 3.7 folds, respectively. On the other hand, TQ significantly downregulates TWIST-1 expression to 18% of the control expression level. PTX did not induce any significant change in the expression level of the SNAIL-1, SNAIL-2, or TWIST-1 genes. Taken together, TQ efficiently diminishes tumor-associated stem cells (Figure 7). 

## 3. Discussion and Conclusion

Breast cancer remains the most common malignancy in females and a leading cause of death worldwide [3,24]. PTX is a cornerstone and commonly used chemotherapeutic drug for the treatment of breast cancer [9,25]. Despite its promising initial clinical response, it might be discontinued due to the emergence of resistance and toxicities [26,27]. TQ is a major active component of the *Nigella sativa* plant, which is commonly used for different medicinal purposes [18,28]. Herein, we are testing the hypothesis that a combination of PTX with TQ can decrease the breast cancer cell resistance to PTX. 

According to our data, TQ alone showed significantly weaker cytotoxic/antiproliferative effects compared to PTX. Apparently, TQ combined with PTX resulted in decreased PTX potency in the form of a slight increase in its IC_50_ values. It was interesting to discover that TQ significantly abolished the resistance fractions of both breast cancer cell lines to PTX (R-fractions were above 40% in both cell lines). In one of our previous publications, it was found that curcuminoid-based synthetic compounds increased the IC_50_ of PTX against colorectal cancer cells but significantly decreased the resistance fractions of these cells towards PTX [16]. Herein, we have a similar scenario with the natural and safe compound, TQ. Cumulative evidence within the literature reports the safety and potency of TQ in enhancing many chemotherapeutic compounds against different tumor cell lines [29,30,31,32]. 

It is well known that PTX causes cell cycle arrest in the G2/M phase [33]. According to our finding in the current study, TQ induced accumulation of cells in S-phase. This could explain the apparent antagonism between PTX and TQ. However, this combination resulted in an increase in the pre-G cell population. Yet, TQ pushed quiescent stem cells to proliferate and become sensitized to PTX treatment via entering S-phase temporally and then entering G2/M-phase [34].

The elevated Pre-G cell population due to PTX treatment alone or in combination with TQ is indicative of cell death. Apoptotic, necrotic, and autophagic cell death induced by PTX, TQ, and their combination were examined to test the above hypothesis. Our observations showed that TQ significantly increased apoptosis in T47D cells by more than 4 folds and 16 fold after 24 h and 48 h, respectively. Many previous publications from our team and from others have highlighted the apoptotic effects of TQ against several cancer cells [31,32,35,36]. The enhancement effect of TQ towards PTX against T47D could be explained by the supra increase of apoptosis; PTX in combination with TQ increased the apoptosis by more than 4 folds compared to PTX alone. It is worth mentioning that combination treatment induced marked elevation rates of necrosis compared to PTX alone. It was previously reported by Canatan and colleagues that the combination of PTX with TQ increased the expression of several genes involved in apoptosis in triple-negative breast cancer [37]. On the other hand, MCF-7 does not undergo normal apoptosis due to a lack of caspase-3 expression [38]. In the current work, caspase-3 was not detected by Western blotting in MCF-7 cells. Yet, it was suggested in our previous publications that autophagy represents an alternative cell death pathway in MCF-7 cells [32]. Herein, TQ induced autophagic pro-death effects in MCF-7 cells. In other words, PTX and TQ increased the concentration of caspase-3 enzyme in T47D cells with a prominent elevation of cleaved PARP concentration, which is indicative of apoptosis-dependent cell death. In MCF-7 (caspase-3 deficient cell line), TQ and PTX forced the cells to proceed in autophagy-dependent cell death. 

Breast cancer stem cells play an important role in resisting chemotherapeutic treatments, and targeting or depleting this clone could effectively sensitize cancer cells to drugs [39]. Overall, our results provide further evidence to the ant-resistance effect exhibited by TQ in combination with PTX through studying their effect against breast cancer-associated stem cells (CD44+/CD24-) [40]. To the best of our knowledge, this is the first study demonstrating the effect of TQ in depleting BCSCs. Herein, TQ remarkably decreased the percentage of the CD44+/CD24- cell clone of MCF-7 and T47D, while PTX caused a slight decrease in the CD44+/CD24- cell clone. Interestingly, TQ in combination with PTX significantly decreased the aforementioned clone far more than TQ or PTX treatment alone. Resistance to chemotherapy and poor cancer patient prognosis was previously attributed to the failure of chemotherapy in depleting this stem cell clone [41]. Finally, the TQ effect in depleting tumor-associated stem cells was confirmed via studying the expression levels of key EMT genes. BCSCs possess high expression levels of mesenchymal markers such as TWIST-1 and a low expression of epithelial markers [42]. Herein, TQ significantly downregulates EMT-regulatory protein (TWIST-1) expression. Previously, it was found that TQ downregulates TWIST-1 expression in other tumor types, resulting in improved efficacy [43]. Other key EMT genes, the SNAIL gene family, are known to promote the migration and invasion of cancer cells [44]. Further details showed that the SNAIL family increased the sensitivity to anti-tubulin drugs such as PTX through the downregulation of bIII and bIVa-tubulin [45]. Thus, the observed increased expression of SNAIL1 and SNAIL2 due to TQ treatment, in the current study, provides PTX with a wider target to induce mitotic catastrophe, resulting in ultimately reduced resistance.

In conclusion, TQ proved and is still proving to have a potential effect in chemosensitizing several tumor types such as breast cancer to many chemotherapeutic agents such as PTX. Several aspects of TQ in decreasing breast cancer cell resistance to PTX are shown in this study. Mechanisms underlying this effect include cell cycle synchronization followed by cell death via apoptosis, necrosis, and autophagy. In addition, TQ interacts with EMT properties resulting in diminishing tumor-associated resistant stem cell fraction. Further in vivo studies that translate these molecular observations into therapeutic values is highly recommended.

## 4. Materials and Methods 

### 4.1. Drugs and Chemicals

Thymoquinone (TQ), paclitaxel (PTX), and sulfarodamine-B (SRB) were all obtained from Sigma-Aldrich Chemical Co. Cell culture media, fetal bovine serum, and trypsin were obtained from Gibco™, Thermo Fisher Scientific.

### 4.2. Cell Culture

MCF-7 and T47D, were obtained from Nawah Scientific (Mokkatam, Cairo, Egypt). Cells were maintained in full DMEM media with heat-inactivated fetal bovine serum (10% *v*/*v*), streptomycin (100 μg/mL), and penicillin (100 units/mL). Cells were kept in a humidified, 5% (*v*/*v*) CO2 atmosphere at 37 °C. 

### 4.3. Cell Viability Assay

An SRB assay was used to evaluate the cytotoxicity effect of TQ, PTX, and/or their combination against MCF7 and T47D as previously described. Cells were seeded at 3000–5000 cells/well and treated with a serial concentration of TQ (0.01–300 µM), PTX (0.001–10 µM), and their combination for 72 h. Following that, cells were fixed by adding TCA (10% *w*/*v*) to each well and incubated for 1 h at 4 °C. After washing, a 0.4% SRB staining solution (*w*/*v*) was added, and the following steps and incubations followed what was previously described. The absorbance was measured at 540 nm with an ELISA microplate reader and calculated as the percent viability of control cells (cells exposed to drug-free media). DMSO concentrations were less than 0.1% in all treatment conditions.

### 4.4. Data Analysis

The dose–response curves of TQ, PTX, and their combination were analyzed using the E_max_ model [46] according to the following formula
(1)$$\%\;\mathrm{Cell}\;\mathrm{viability}=\left(100-R\right)\times \left(1\;-\frac{{\left[D\right]}^{m}}{K_{d}{}^{m}+\;{\left[D\right]}^{m}\;}\right)+R$$
The CI-value was calculated from the following formula:

Combination index (CI) was calculated from the formula: (2)$$CI=\frac{\mathrm{IC}50\;\mathrm{of}\;\mathrm{drug}\left(x\right)\;\mathrm{combination}}{\mathrm{IC}50\;\mathrm{of}\;\mathrm{drug}\left(x\right)\;\mathrm{alone}}+\frac{\mathrm{IC}50\;\mathrm{of}\;\mathrm{drug}\left(y\right)\;\mathrm{combination}}{\mathrm{IC}50\;\mathrm{of}\;\mathrm{drug}\left(y\right)\;\mathrm{alone}}$$

The nature of the drug interaction was defined according to Chou and Talalay as synergism if CI < 0.8, as antagonism if CI > 1.2, and as additive if CI ranges from 0.8 to 1.2 [47]. 

### 4.5. Cell Cycle Analysis by Flow Cytometry

To evaluate the effect of the drugs on cell cycle distribution, both cell lines were treated by the pre-determined IC_50_ values of TQ, PTX, or both drugs in combination, for 24 or 48 h. An additional drug-free medium-treated group acted as a control group. Post-treatment, cells were trypsinized, collected, and washed with ice-cold PBS and re-suspended in 0.5 mL of PBS. To ensure fixation of cells, 2 mL of 60% ice-cold ethanol were added on the cells while vortexing. Cells were incubated at 4 °C for 1 h. Prior to analysis, cells were washed and re-suspended in 1 mL of PBS containing 50 µg/mL RNAase A and 10 µg/mL propidium iodide (PI). Cells were incubated for 20 min in the dark at 37 °C and analyzed for DNA content using flow cytometry analysis FL2 (λex/em 535/617 nm). In all of the following flow cytometer analysis, 12,000 events were acquired, and NovoExpress™ software was used for analysis.

### 4.6. Analysis of Cell Apoptosis by Flow Cytometry

An Annexin V-FITC apoptosis detection kit (Abcam Inc., Cambridge Science Park, Cambridge, UK) was used to determine the effect of drugs on apoptosis and necrosis. Briefly, the cells were treated by the pre-determined IC_50_ values of either TQ, PTX, or both drugs combined for 24 h. A drug-free media-treated group was used as control. Cells were collected and washed twice with PBS and incubated in a dark place with 0.5 mL of Annexin V-FITC/PI solution for 30 min at room temperature according to the manufacturer’s protocol. FITC and PI fluorescent signals were then analyzed using FL1 and FL2 signal detector, respectively (λex/em 488/530 nm for FITC and λex/em 535/617 nm for PI). Positive FITC and/or PI cells were quantified by quadrant analysis. 

### 4.7. Analysis of Cell Autophagy by Flow Cytometry 

To further confirm the cell death mechanism induced by the drugs, autophagic cell death was quantitatively analyzed using a Cyto-ID Autophagy Detection Kit (Abcam Inc., Cambridge Science Park, Cambridge, UK). In brief, cells were treated for 24 h by the IC_50_ values of the test compounds (single or combined treatments). Chloroquine treatment (10 µM) was used as a positive control, while a drug-free medium was used as a negative control. Cells were then washed twice with PBS and stained with Cyto-ID Green in the dark at 37 °C for 30 min according to the manufacturer’s protocol. After staining, cells were analyzed for Cyto-ID differential green/orange fluorescent signals using an FL2 signal detector (λex/em 535/617 nm). Mean net fluorescent intensities (NFI) were quantified.

### 4.8. Stem Cell Detection by Flow Cytometry 

For assessing the effects of TQ, PTX, and their combination against the breast cancer-associated stem cell clone (CD44+/CD24-), cells underwent FITC-labeled anti-CD44 and APC/Cy7-labeled anti-CD24 antibody (Abcam Inc. Cambridge Science Park, Cambridge, UK) staining and flowcytometry assessment. Briefly, cells were treated with the predetermined IC_50_ values of test drugs (single or combined treatments), or with drug-free media as a control group, for 24 h. After treatment, cells were trypsinized and washed with ice-cold PBS supplemented with 10% FCS. Cells were stained with the conjugated anti-CD44 and anti-CD24 antibodies and kept in the dark at room temperature for 30 min. Cells were then washed three times with ice-cold PBS containing 10% FCS. Cells were then analyzed for FITC and APC/CY7 fluorescent signals using FL1 and FL2 signal detector, respectively (λex/em 488/530 nm for FITC and λex/em 535/617 nm for APC/CY7). 

### 4.9. EMT Gene Expression Analysis

Real-time polymerase chain reaction (PCR) was performed to assess the expression of CDH1, CDH2, SNAL1, SNAL2, ZEB2, and TWIST1 genes after treatment with the pre-determined IC_50_ values of TQ and PTX. After 24 h of treatment, RNA was extracted using mirVana™ RNA isolation kit (Invitrogen, Carlsbad, CA, USA). The RNA and purity were confirmed (A260/280>2.0) using a DeNovix DS-11™ microvolume spectrophotometer (Thermo Fisher Scientific, Wilmington, DE, USA). Subsequently, the total RNA samples of all treatments were reverse-transcribed to construct a cDNA library using the SuperScript™ Master Mix kit (Invitrogen, Carlsbad, CA, USA). The cDNA were then subjected to quantitative real-time PCR reactions using Custom TaqMan^®^ Gene Expression Assay (Applied Biosystems, Foster City, CA, USA). GAPDH was used as a housekeeping gene; normalized fold changes for all genes of interest were calculated using the following formula: 2-ΔΔCq.

### 4.10. Autophagy Gene Expression Analysis

Real-time polymerase chain reaction (PCR) was performed on the cDNA prepared in the previous experiment to assess the expression of beclin-1 and LC3-II autophagy genes after treatment with the pre-determined IC_50_ values of TQ and PTX. The beclin-1 forward primer was 3’-GGCTGAGAGACTGGATCAGG-5’; the backward primer was 5’- CTGCGTCTGGGCATAACG-3’; the LC3-II forward primer was 3’-GAGAAGCAGCTTCCTGTTCTGG-5’; the backward primer was 5’-GTGT CCGTTCACCAACAGGAAG-3’. The housekeeping β-actin gene was as a reference gene with a forward primer of 3’-GAGAGGCGGCTAAGGTGTTT-5’ and a backward primer of 5’-TGGTGTAGACGGGGATGACA-3’.

### 4.11. Western Blot Analysis and Detection of Apoptosis Related Signals

Apoptosis proteins, caspase-3, and PARP were assessed within cell lysate after treatment with TQ, PTX, and their combination to confirm cell death propagation via apoptosis. Briefly, cells were treated with the pre-determined IC_50_ values of PTX and TQ for 24 h and 48 h. Cell lysates were extracted using an RIPA-buffer and electrophoresed using SDS-PAGE (10%) and then transferred to PVDF-membrane. Caspase-3 and cleaved PARP-1 proteins were detected using rabbit monoclonal anti-active caspase-3 and rabbit monoclonal anti-PARP (Abcam Inc., Cambridge Science Park, Cambridge, UK). Bands were visualized using HRP-conjugated anti-rabbit secondary antibodies (Abcam Inc., Cambridge Science Park, Cambridge, UK). 

### 4.12. Statistical Analysis

Data are presented as mean ± SD using Prism^®^ for Windows, ver. 5.00 (GraphPad Software Inc., La Jolla, CA, USA). To assess significance, analysis of variance (ANOVA) with an LSD post hoc test was used with SPSS^®^ for Windows, version 17.0.0. A cut off value of *p* < 0.05 was used for significance.

## Figures and Tables

**Figure 1 molecules-25-00426-f001:**
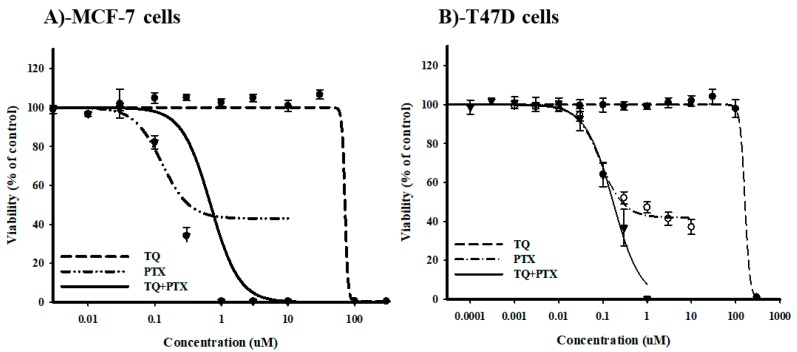
The effect of thymoquinone (TQ) on the dose-response curve of paclitaxel (PTX) in MCF-7 (**A**) and T47D (**B**) breast cancer cell lines. Cells were exposed to the serial dilution of PTX, TQ, or their combination for 72 h. Cell viability was determined using a sulfarodamine-B (SRB) assay, and data are expressed as mean ± SD (*n* = 3).

**Figure 2 molecules-25-00426-f002:**
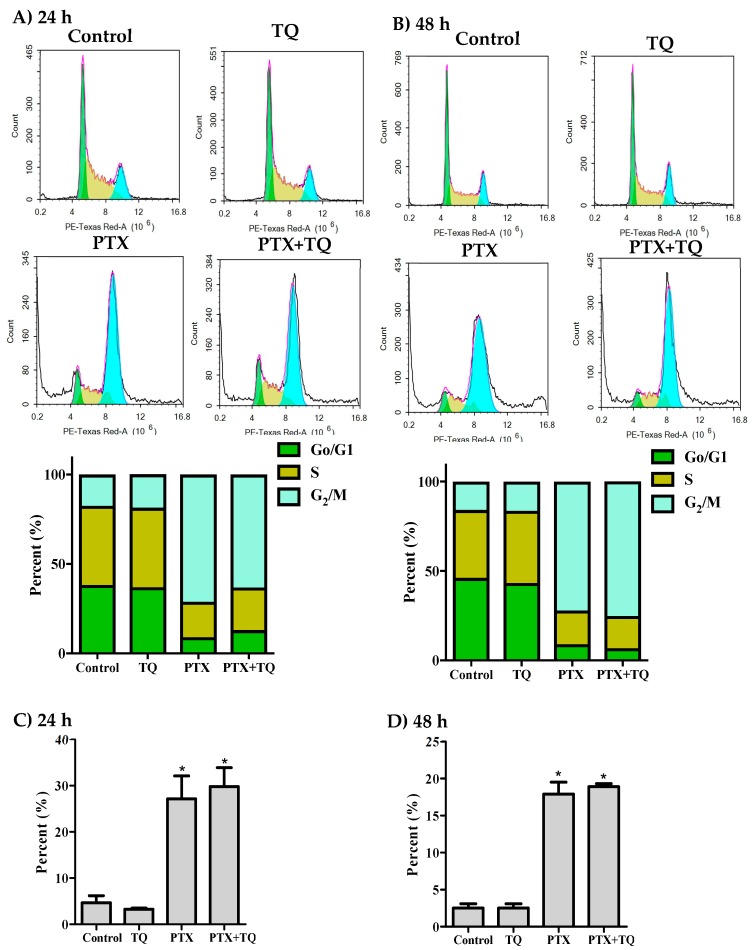
Effect of PTX, TQ, and their combination on the cell cycle distribution of MCF-7 cells. Cells were exposed to PTX, TQ, or their combination for 24 h (**A**,**C**) or 48 h (**B**,**D**). Cell cycle distribution was determined using DNA content flowcytometry analysis and different cell phases were plotted as percentage of total events. Sub-G cell population was plotted as percent of total events (**C**,**D**). Data are presented as mean ± SD; *n* = 3. (*) significantly different from the control group.

**Figure 3 molecules-25-00426-f003:**
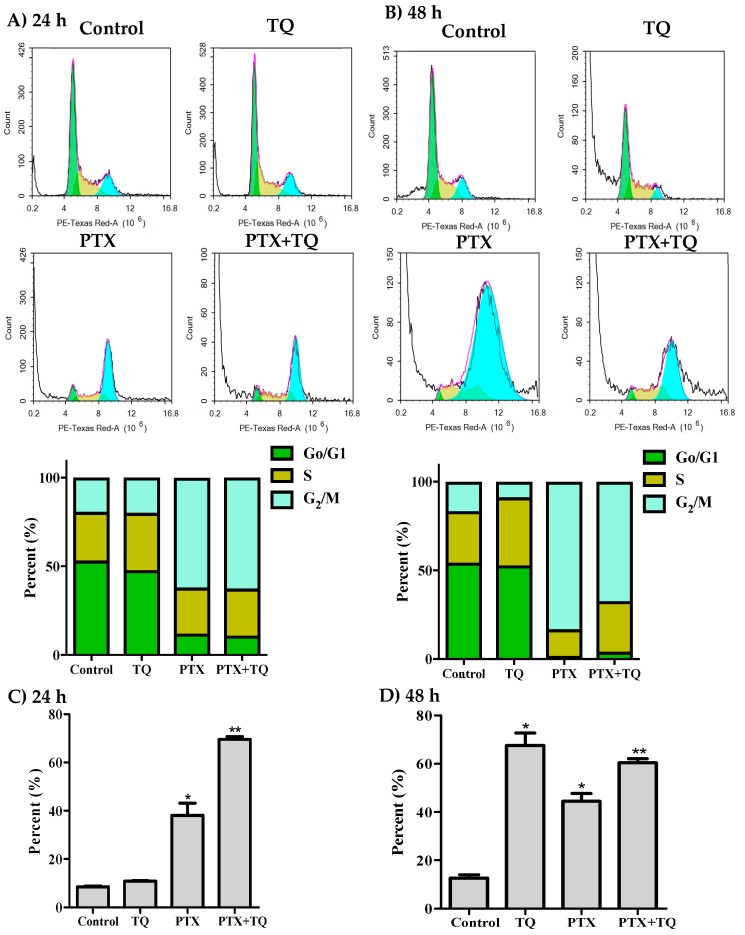
Effect of PTX, TQ, and their combination on the cell cycle distribution of T47D cells. Cells were exposed to PTX, TQ, or their combination for 24 h (**A**,**C**) or 48 h (**B**,**D**). Cell cycle distribution was determined using DNA content flowcytometry analysis and different cell phases were plotted as percentage of total events. Sub-G cell population was plotted as percent of total events (**C**,**D**). Data are presented as mean ± SD; *n* = 3. (*) significantly different from the control group. (**) significantly different from PTX treatment.

**Figure 4 molecules-25-00426-f004:**
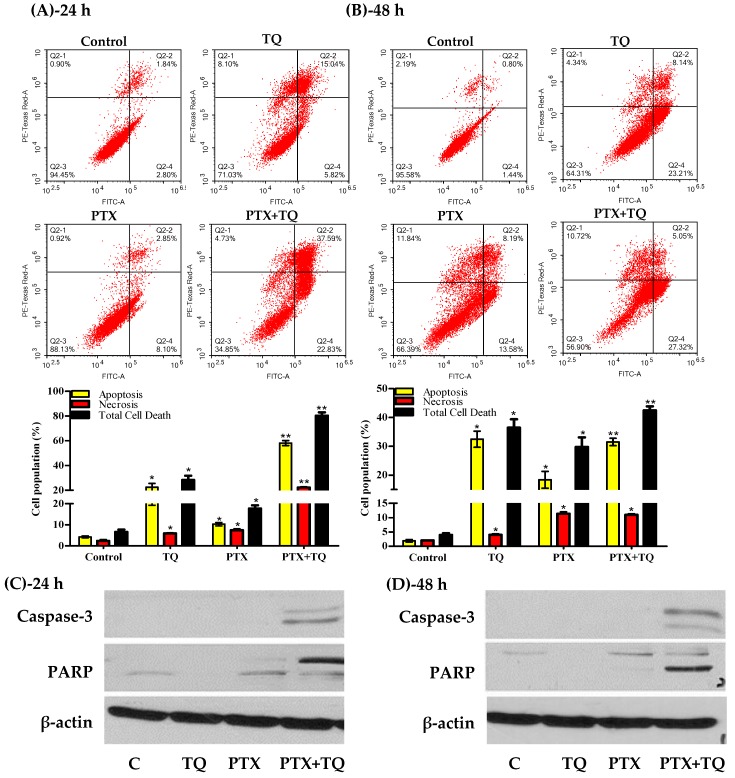
Apoptosis/necrosis assessment in T47D cells after exposure to PTX, TQ, and their combination. Cells were exposed to PTX, TQ, or their combination for 24 h (**A**) and 48 h (**B**). Cells were stained with annexin V-FITC/PI and different cell populations are plotted as a percentage of total events. Western blot analysis for caspase-3 and PARP was assessed for MCF-7 (**C**) and T47D (**D**) cells. Data are presented as mean ± SD; *n* = 3. (*) significantly different from the control group. (**) significantly different from PTX treatment.

**Figure 5 molecules-25-00426-f005:**
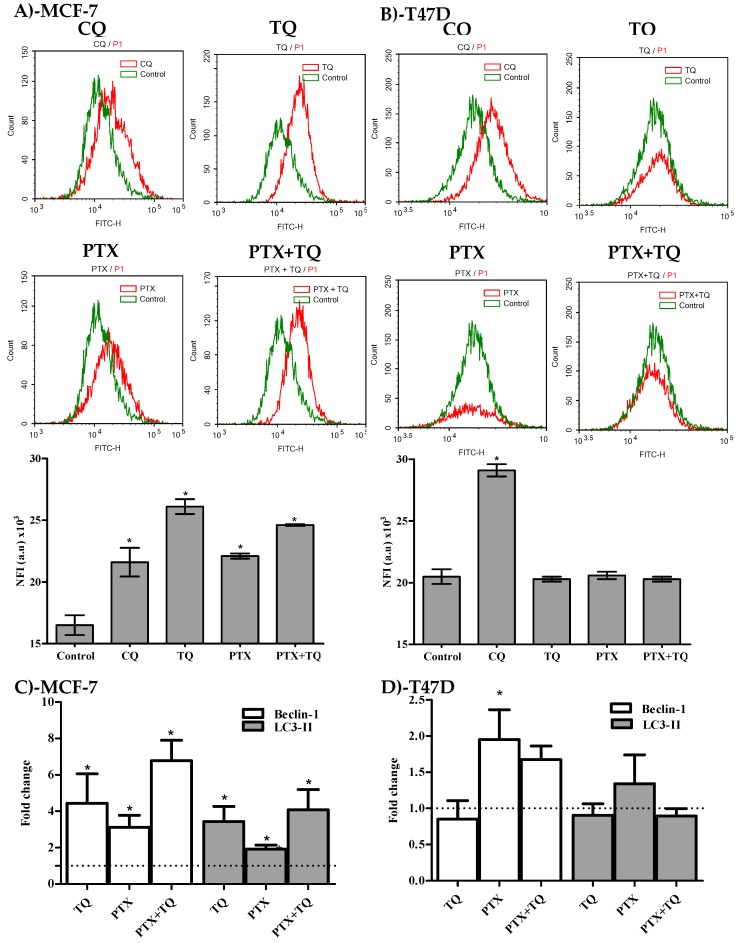
Autophagic cell death assessment in MCF-7 (**A**) and T47D (**B**) cells after exposure to PTX, TQ, and their combination. Cells were exposed to PTX, TQ, or their combination for 24 h, and were stained with a Cyto-ID autophagosome tracker. Net fluorescent intensity (NFI) was plotted and compared to the basal fluorescence of the control group. Gene expression fold changes for beclin-I and LC3-II were assessed for MCF-7 (C) and T47D cells (**D**). Data are presented as mean ± SD; *n* = 3. (*) significantly different from the control group.

**Figure 6 molecules-25-00426-f006:**
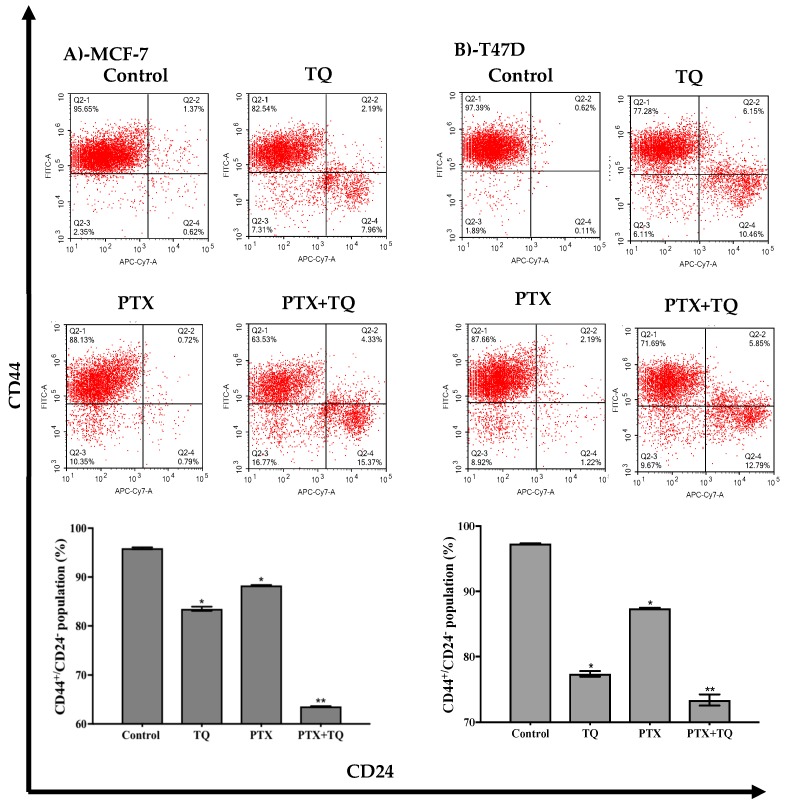
Effect of PTX, TQ, and their combination on the expression of CD44 and CD24 stem cell markers. MCF-7 (**A**) and T47D (**B**) cells were exposed to PTX, TQ, or their combination for 24 h. Expression levels of CD44 and CD24 were assessed using flowcytometry and plotted as percentage of total events. Data are presented as mean ± SD; *n* = 3. (*) significantly different from the control group. (**) significantly different from PTX treatment.

**Figure 7 molecules-25-00426-f007:**
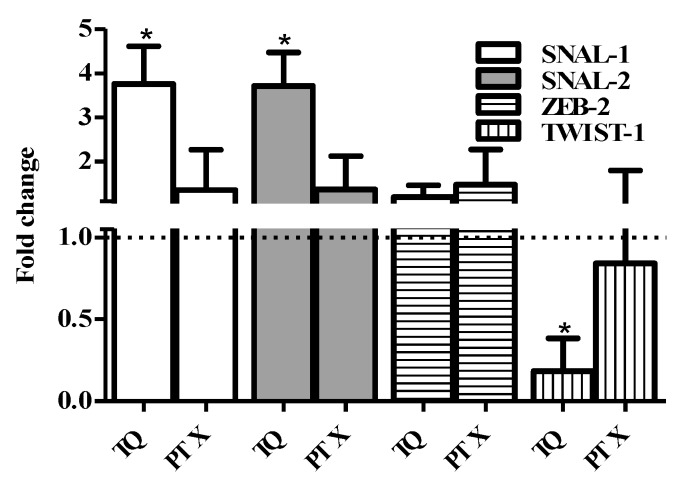
Effect of PTX and TQ on the expression of EMT-related genes in MCF-7 cells. Cells were exposed PTX or TQ for 24 h. Total RNA was extracted and subjected to RT-qPCR to measure gene expression. Data were plotted using the 2-ΔΔCt method (expression normalized to the housekeeping gene GAPDH). Fold expression and significance was calculated relative to control untreated cells (dotted line). Data are presented as mean ± SD; *n* = 3. (*) significantly different from the control group.

**Table 1 molecules-25-00426-t001:** Combination analysis of cell cytotoxicity for TQ, PTX, and their combination against MCF-7 and T47D breast cancer cell lines.

	MCF-7		T47D	
	IC50 (µM)	R-Value (%)	IC50 (µM)	R-Value (%)
PTX	0.2 ± 0.07	42.3 ± 1.4	0.1 ± 0.01	41.9 ± 1.1
TQ	64.9 ± 14.5	1.6 ± 1.3	165.1 ± 2.8	0.1 ± 0.15
PTX+TQ	0.7 ± 0.01	0	0.15 ± 0.02	0
CI-value	Antagonism/4.6		Antagonism/1.6	

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
