# Peer review of "Thymoquinone Enhances Paclitaxel Anti-Breast Cancer Activity via Inhibiting Tumor-Associated Stem Cells Despite Apparent Mathematical Antagonism"

_molecules, 2020, doi:10.3390/molecules25020426_

Round 1

Reviewer 1 Report

The manuscript entitled "enhances paclitaxel anti-breast cancer activity via inhibiting tumor-associated stem cells despite apparent mathematical antagonism" reports number of functional and bioassays to prove its combination anticancer activity. The data is supportive to their conclusion. However, the conclusion needs further assays such as western blot for visual confirmations. Additionally, I strongly recommend to prove this in animal model otherwise the data validity is some what poor.

Author Response

Reviewer # 1

1) The manuscript entitled "enhances paclitaxel anti-breast cancer activity via inhibiting tumor-associated stem cells despite apparent mathematical antagonism" reports number of functional and bioassays to prove its combination anticancer activity. The data is supportive to their conclusion. However, the conclusion needs further assays such as western blot for visual confirmations. Additionally, I recommend to prove this in animal model otherwise the data validity is somewhat poor.

RE: We appreciate the reviewer’s comment.

Accordingly, western blot confirmation for apoptosis key markers (caspase-3 and PARP) has been done and abandoned to the revised version (revised Figure 5-C&D).

2) Additionally, I recommend to prove this in animal model otherwise the data validity is somewhat poor.

RE: We appreciate the reviewer’s comment and we value the animal experiment for proving the activity. However, the time allowed for the revision does not permit carrying-out the in-vivo experiment. Besides, the major aim of the current study is to correlate the mathematical resistance within two different breast cancer cell lines with the abundance of tumor-associated stem cells. In other words, this paper is among a series of publications for our team[1] regarding the molecular investigational of TQ rather than discovering the therapeutic impact of TQ and PTX. Our team already published one study regarding the in-vivo impact of TQ and Nigella sativa extract[2] and other publications in this line of research is under preparation

Accordingly and in order to respond to the reviewer’s comment, a clear statement for the further in-vivo study is mentioned in the discussion section of the revised version.

[1] Bashmail HA, Alamoudi AANoorwali A, Hegazy GA, AJabnoor G, Choudhry H, Al-Abd AM. Thymoquinone synergizes gemcitabine anti-breast cancer activity via modulating its apoptotic and autophagic activities. Sci Rep. 2018;8(1):11674. doi: 10.1038/s41598-018-30046-z.

Alaufi OM, Noorwali A, Zahran F, Al-Abd AM, Al-Attas S. Cytotoxicity of thymoquinone alone or in combination with cisplatin (CDDP) against oral squamous cell carcinoma in vitro. Scientific Report 2017;7(1):13131. doi: 10.1038/s41598-017-13357-5

[2] Al-Attas SA, Munshi A, Noorwali A, Algrigri MA, Abohager EA, Zahran FM. Nigella Sativa Extract Chemoprevention in Oral Cancer:in vivoStudy. Advances inEnvironmental Biology. 2015. 9(22): 75-90

Reviewer 2 Report

1. This manuscript needs to supply some experiments on genes or proteins related to autophagy and EMT. Does breast cancer stem cells (BCSC) exist in breast cancer cells (MCF-7 and T47D cells)? We hope that the author explain the relationship between pre-G cell and BCSC in the discussion. In the discussion, MCF-7 cells do not undergo normal apoptosis due to lack of caspase-3 expression. Which pathway does TQ promote MCF-7 cells apoptosis? If conditions permit, we hope that the author can supply an animal experiment to further explain that TQ enhances PTX.

Author Response

Reviewer #2:

This manuscript needs to supply some experiments on genes or proteins related to autophagy and EMT.

RE: We appreciate the reviewer’s comment.

Accordingly and in order to respond to the reviewer’s comment, we confirmed autophagy by further assessing the expression of two key markers for autophagy (beclin-1 and LC3-II genes) in both cell lines and the new data can be found in the revised Figure-4C&D. Also, additional western blot analysis was carried out for apoptosis key markers (caspase-3 and PARP) in T47D cells and the new data can be found in the revised Figure 5-C&D.

Does breast cancer stem cells (BCSC) exist in breast cancer cells (MCF-7 and T47D cells)? We hope that the author explain the relationship between pre-G cell and BCSC in the discussion.

RE: We appreciate the reviewer’s comment. Yes, BCSC exists in both MCF-7 and T47D cells according to our and other teams’ publications[1]. Pre-G cells indicate cell fragments (unspecified cell death) which could reflect programmed cell death (apoptosis, autophagy or other mechanisms) or non-programmed cell death such as necrosis. This is why we further dissected the potential mechanisms of cell death in both cell lines under investigation.

Accordingly and to avoid confusion, further clarification for Pre-G phase cells was included in the results section.

In the discussion, MCF-7 cells do not undergo normal apoptosis due to lack of caspase-3 expression. Which pathway does TQ promote MCF-7 cells apoptosis?

RE: We appreciate the reviewer’s scientific accuracy; according to our observation, TQ promotes autophagy-dependent cell death due to deficiency of apoptosis machinery (caspase enzymes) in MCF-7 cells.

Accordingly and in order to respond to the reviewer’s comment, an additional explanation for this point has been further discussed in the discussion section.

If conditions permit, we hope that the author can supply an animal experiment to further explain that TQ enhances PTX.

RE: We appreciate the reviewer’s comment. However, the time allowed for the revision does not permit carrying-out the in-vivo experiment. Besides, the major aim of the current study is to correlate the mathematical resistance within two different breast cancer cell lines with the abundance of tumor-associated stem cells. In other words, this paper is among a series of publications for our team[2] regarding the molecular investigational of TQ rather than discovering the therapeutic impact of TQ and PTX.

[1] Bashmail HA, Alamoudi AANoorwali A, Hegazy GA, AJabnoor G, Choudhry H, Al-Abd AM. Thymoquinone synergizes gemcitabine anti-breast cancer activity via modulating its apoptotic and autophagic activities. Sci Rep. 2018;8(1):11674. doi: 10.1038/s41598-018-30046-z.

Akrap N, Andersson D,Bom E, Gregersson P, Ståhlberg A, and Landberg G. Stem Cell Reports. Identification of Distinct Breast Cancer Stem Cell Populations Based on Single-Cell Analyses of Functionally Enriched Stem and Progenitor Pools 2016 12; 6(1): 121–136

[2] Bashmail HA, Alamoudi AA, Noorwali A, Hegazy GA, AJabnoor G, Choudhry H, Al-Abd AM. Thymoquinone synergizes gemcitabine anti-breast cancer activity via modulating its apoptotic and autophagic activities. Sci Rep. 2018;8(1):11674. doi: 10.1038/s41598-018-30046-z.

Alaufi OM, Noorwali A, Zahran F, Al-Abd AM, Al-Attas S. Cytotoxicity of thymoquinone alone or in combination with cisplatin (CDDP) against oral squamous cell carcinoma in vitro. Scientific Report 2017;7(1):13131. doi: 10.1038/s41598-017-13357-5

Reviewer 3 Report

This work deals with experiments designed to assess the suitability of paclitaxel/thymoquinone against brest cancer models.

The work appears carried out competently  and the outcome is of interest for the journal's readership. On the other hand the novelty is not very high both in terms of experiments and approach that are quite routinian. However, the methods and results here reported are well described and can be of relevance for people involved in similar research and more in general in cancer research. Additionally, the conclusions and the discussion are well supported by the experimental evidences that have been critically esamined. Overall, I suggest the publication of this work after a proper revision. Details are included in the attached files (comments) and a typos/english check is necessary in the whole text. Also, the authors should be pay attention to the quality of the submitted paper in terms of misprints an residual sentences included in the template that should be removed before submission.

Author Response

Reviewer #3:

This work deals with experiments designed to assess the suitability of paclitaxel/thymoquinone against breast cancer models.

The work appears carried out competently and the outcome is of interest for the journal's readership. On the other hand the novelty is not very high both in terms of experiments and approach that are quite routine. However, the methods and results here reported are well described and can be of relevance for people involved in similar research and more in general in cancer research. Additionally, the conclusions and the discussion are well supported by the experimental evidences that have been critically examined. Overall, I suggest the publication of this work after a proper revision. Details are included in the attached files (comments) and a typos/English check is necessary in the whole text. Also, the authors should be pay attention to the quality of the submitted paper in terms of misprints an residual sentences included in the template that should be removed before submission.

RE: We appreciate the reviewer’s detailed comments. All comments in the attached pdf file were followed.

From experimental section it appears that these experiments were carried out at 24 h and 48 h.

RE: We appreciate the reviewer’s comment. Theoretically, it is correct. However, according to our previous experience with similar experiments[1], several early events of apoptosis, cell cycle and autophagy cannot be detected at the terminal cell death stage after 72 h. Events like cell cycle arrest, early apoptosis, and early autophagic vacuoles are missed if we carry out these experiments at the same time points defined by the SRB-assay. The majority of the remaining cells are debris, late apoptosis, and necrotic bodies.

Several lines from the template submission file (last 3 lines of the discussion and last paragraphs of the materials and methods) were missed within the text of the manuscript.

RE: We appreciate the reviewer’s accuracy. The manuscript was double-checked and missed template lines were deleted

The conclusion is recommended to be merged with the discussion and the section is to be named “discussion and conclusion”.

RE: We appreciate the reviewer’s comment. The conclusion and discussion sections were merged.

[1] Henidi HA, Al-Abd AM, Al-Abbasi FA, BinMahfouz HA, El-Deeb IM. Design and synthesis of novel phenylaminopyrimidines with antiproliferative activity against colorectal cancer. RSC Adv. 2019; 9: 21578 doi: 10.1039/c9ra03359a

Fekry MI, Ezzat SM, Salama MM, Alshehri OY, Al-Abd AM. Bioactive glycoalkaloids isolated from Solanum melongena fruit peels with potential anticancer properties against hepatocellular carcinoma cells. Sci Rep. 2019;9(1):1746. doi: 10.1038/s41598-018-36089-6

Bashmail HA, Alamoudi AA, Noorwali A, Hegazy GA, AJabnoor G, Choudhry H, Al-Abd AM. Thymoquinone synergizes gemcitabine anti-breast cancer activity via modulating its apoptotic and autophagic activities. Sci Rep. 2018;8(1):11674. doi: 10.1038/s41598-018-30046-z.

Baghdadi MA, Al-Abbasi FA, El-Halawany AM, Aseeri AH, Al-Abd AM. Anticancer profiling for coumarins and related O-naphthoquinones from Mansonia gagei against solid tumor cells in-vitro. Molecules 2018; 23(5). pii: E1020. doi: 10.3390/molecules23051020.

Round 2

Reviewer 1 Report

Authors responded enough.

Reviewer 2 Report

Accept in its current form.